# Role of Niobium on the Passivation Mechanisms of TiHfZrNb High-Entropy Alloys in Hanks’ Simulated Body Fluid

**DOI:** 10.3390/jfb15100305

**Published:** 2024-10-14

**Authors:** Ayoub Tanji, Xuesong Fan, Ridwan Sakidja, Peter K. Liaw, Hendra Hermawan

**Affiliations:** 1Department of Mining, Metallurgical, and Materials Engineering, Laval University, Quebec City, QC G1V 0A6, Canada; ayoub.tanji@cegeptr.qc.ca; 2Department of Materials Science and Engineering, The University of Tennessee, Knoxville, TN 37996, USA; xfan5@vols.utk.edu (X.F.); pliaw@utk.edu (P.K.L.); 3Department of Physics Astronomy and Materials Science, Missouri State University, Springfield, MO 65897, USA; ridwansakidja@missouristate.edu

**Keywords:** biomaterials, electrochemical impedance spectroscopy, high-entropy alloy, polarization, modeling, passivation

## Abstract

A family of TiHfZrNb high-entropy alloys has been considered novel biomaterials for high-performance, small-sized implants. The present work evaluates the role of niobium on passivation kinetics and electrochemical characteristics of passive film on TiHfZrNb alloys formed in Hanks’ simulated body fluid by analyzing electrochemical data with three analytical models. Results confirm that higher niobium content in the alloys reinforces the compactness of the passive film by favoring the dominance of film formation and thickening mechanism over the dissolution mechanism. Higher niobium content enhances the passivation kinetics to rapidly form the first layer, and total surface coverage reinforces the capacitive-resistant behavior of the film by enrichment with niobium oxides and reduces the point defect density and their mobility across the film, lowering pitting initiation susceptibility. With the high resistance to dissolution and rapid repassivation ability in the aggressive Hanks’ simulated body fluid, the TiHfZrNb alloys confirm their great potential as new materials for biomedical implants and warrant further biocompatibility testing.

## 1. Introduction

The fabrication of biomedical implants and devices has mostly relied on high corrosion-resistant alloys, such as Co-28Cr-6Mo, Ti-6Al-4V, and stainless-steel type 316L. The stable passive film formed on the surface of the alloys prevents the release of metal ions into the surrounding implantation site, ensuring their good corrosion resistance and, therefore, their biocompatibility [1,2]. Prior to approval by regulatory organizations, implant materials are required to show high resistance to corrosion in simulated body fluids (in vitro) to anticipate inertness in the human body (in vivo) [3,4]. Among the many new materials recently proposed for implant materials, high-entropy alloys (HEAs) stand out with their excellent mechanical properties [5,6,7], making them very attractive for high-performance small-size implants, such as coronary stents and reduced-diameter dental implants. The diverse composition of HEAs allows the use of non-toxic materials to guarantee biocompatibility and provides a tailored mechanical property adaptable to the specific needs of different types of implants [8,9,10]. The TiZrHfNb-based families are among the most studied HEAs that show a high corrosion resistance in chloride-containing simulated body fluids [11,12,13]. One of the reasons for the outstanding corrosion behavior is the formation of a passive layer at the metal surface, often associated with the presence of Ti species. In our previous work, we highlighted the enhancement of protectiveness related to increased Nb content and a high corrosion-resistant passive film of TiHfZrNb in Hanks’ simulated body fluid at 37 °C has been confirmed [14], but their detailed passivation mechanism is yet to be revealed. On the other hand, we confirmed that Nb is beneficial for corrosion resistance by stabilizing the oxide layer.

The corrosion resistance of an alloy is strongly associated with a self-passivation behavior, which reflects its passivation capacity and film stability [15]. For this purpose, an ideal biomedical alloy will have a stable passive film that resists corrosion attack in the aggressive physiological environment and a high capacity for repassivation and damage correction [16]. This capacity is linked to the kinetic properties of a passive film, its semiconducting properties, and its electronic structure [17,18]. Several models have been proposed and used for studying passive film kinetics. Two analytical models, the high-field ionic conduction model and the anodic dissolution and film growth model, have been used to analyze and follow different mechanisms and processes during passive film formation and growth on a metal surface [19,20,21,22]. In addition, the point defect model can explain the passivation/repassivation mechanism as a result of point defect migration and diffusion under the influence of an electrostatic field between two film interfaces [23]. The point defects diffusivity in a passive film can become the key to outlining the different growth and degradation mechanisms in the transient- and steady-state [24], and hence, the kinetic properties of the film, as they are strongly associated with the density and diffusivity of defects through the film [25].

The present work aims to reveal the role of niobium (Nb) on the passive film formation and growth process on the surface of three TiHfZrNb alloys by analyzing the electrochemical data obtained from polarization, electrochemical impedance spectroscopy, and Mott–Schottky tests combined with three analytical models: the high-field ionic conduction model, the anodic dissolution and film growth model, and the point defect model. The results of the present work will deepen our comprehension of the role of alloying elements on the passivation mechanism that may help in optimizing the compositional design of HEAs, which results in excellent passive film properties.

## 2. Materials and Methods

An electrical arc-melting process was used to synthesize three compositions of TiZrHfNb_x_ alloys (with x = 0.2, 0.3, and 0.4 in molar ratio and referred to as 0.2 Nb, 0.3 Nb, and 0.4 Nb specimens for simplicity) from a mixture of pure elements, as detailed in previous work [14]. Having an exposed surface of 1 cm^2^, the specimens were mechanically polished with SiC paper up to 800 grit, 6 μm, 1 μm polycrystalline diamond, and 0.02 μm colloidal silica to remove the existing naturally formed oxide and to have a mirror-like appearance and smooth surface. A Hanks’ simulated body fluid (Gibco HBSS(1×), Ref #14,175-095, Fisher Scientific, Nepean, ON, Canada) was used as the electrolyte, having a pH of 7.2 and kept at 37 ± 1 °C throughout the tests.

A set of electrochemical evaluations was performed using a potentiostat (CH Instruments 760E, Austin, TX, USA) in a three-electrode cell set-up with an electrolyte volume of 100 mL. The HEAs specimen was the working electrode, a graphite rod was the counter electrode, and a saturated calomel electrode (SCE) was the reference electrode (hence, all measured potentials are stated as V/SCE). Three tests were performed to gather data: potentiodynamic polarization (PDP), potentiostatic polarization (PSP), and electrochemical impedance spectroscopy (EIS). The evaluation began with PDP tests using a scan rate of 1 mV·s^−1^ after 6 h of stabilization time, and the resulting data were used to identify the passive domain from which a passive polarization potential (*E_pass_*) was selected for the subsequent PSP tests. Before conducting the PSP tests at *E_pass_* for 1 h, a preliminary PSP at *E_corr_* was performed for 1 h. To evaluate the formed passive film at different passivating potentials, EIS tests were performed after the PSP in a frequency range of 100 kHz to 0.01 Hz with an amplitude of 0.01 V at four different passive potentials: −0.1, −0.05, 0, and 0.05 V/SCE. An equivalent electrical circuit was used to simulate the EIS experimental data, which consists of a series and parallel connection of resistors and a constant phase element (CPE). This seems more reasonable to use than a pure capacitance as it compensates for the frequency dispersion resulting from geometrical irregularities, surface roughness, porosity, or resistivity variation across the film thickness. The Mott–Schottky technique, used to characterize the semiconductor properties of passive film, was performed in the passive potential range at a fixed frequency of 1 kHz in the cathodic direction using 50 mV/s scan rate. The CH Instruments and EC-Lab version 10.4 (BioLogic, Seyssinet-Pariset, France) software were used to collect and analyze the electrochemical data.

Three analytical models were used to analyze the passivation mechanism. Firstly, the high-field ionic conduction model (HFM), this model assumes that a passive film is formed and thickened because of a displacement and transfer of charges across the film. This displacement takes the form of ionic conduction resulting from a high-electrical field [26,27,28]. Hence, there is a process of film formation and growth that consumes the charges with 100% yield, with no losses linked to the anodic dissolution. The linear relationship of the current density (*i*) and time (*t*) [26], and between the *i* and surface charge density (*q*) [19], are based on the following equations:(1)$$i=At^{-n}$$
(2)$$\log i\;\left(t\right)=\log A-n\log\left(t\right)\;$$
(3)$$\log i\left(t\right)=\log A+\frac{cBV}{q\left(t\right)}$$
where *A* = constant, and *n* = passivation index that can indirectly express the rate of passive film growth, *c* = constant related to the film, *A* and *B* = parameters associated with the activation energy for the charge displacement, and *V* = potential drop across the film. The second model is the anodic dissolution and film growth model (DGM), at the early period of passivation, the current transient contributes to both passivation and dissolution [22]. Thus, the logarithmic relation of the HFM approach is no longer usable, and the plateau in the logarithmic representation of current density vs. time deviates from the high-field behavior [28,29]. During the phase of the transient current at the early time, the total current generated is dominated by the anodic dissolution process of the metal of which just a small fraction is a result of the film formation process. It is important not to neglect the contribution of the dissolution process in the total current during decay transients at the early time, whereas an alternative mathematical expression was proposed [22]. This approach is not in contradiction with the HFM, but it accommodates all the current sources and the different processes that may take place at the early time, and sequentially separate them to assess their kinetics individually, as expressed by:(4)$$i_{tot}=i_{diss}+i_{film}=\left(1-\theta \right)i_{peak}+\theta At^{-b}$$
(5)$$\theta =1-\exp\left(-kt^{n}\right)$$
where *i_peak_* = peak current, *A* = characteristic charge density coefficient, and *b* = constant, θ = fraction of the film coverage on the surface as described by Avrami kinetics [30], *k* = film growth rate, and *n* = exponent related with the reaction order of the growth mechanism (1D, 2D, and 3D growth) and the geometry of the oxide (spheres, rods, etc.). Finally, according to the point defect model (PDM) theory, the film growth process is an outcome of the diffusivity and flux of oxygen vacancies and/or cationic interstitials across the passive film. For titanium oxides, the dominant point defects in the film are oxygen vacancies that act as electron donors [25,31,32]. The relationship between the donor density and film formation potential can be described by [33]:(6)$$N_{D}=w_{1}.\exp\left(-bE\right)+w_{2}$$
where *w*_1_, *w*_2_, and *b* = unknown constants that can be determined from the experimental data. The diffusivity of point defects (*D*_0_) is expressed based on the Nernst–Planck transport equation [34]:(7)$$w_{2}=\frac{i_{ss}RT}{4eF\varepsilon_{l}D_{0}}$$
(8)$$D_{0}=\frac{i_{ss}RT}{4eF\varepsilon_{l}w_{2}}$$
where *i_ss_* = steady-state current density measured from the PSP curves; *R* = ideal gas constant, *ε_l_* = electric field intensity derived from the linear relationship between the film thickness (*L_ss_*) and *E*, as follows:(9)$$L_{ss}=\frac{\left(1-\alpha \right)E}{\varepsilon_{l}}+B$$
where *α* = polarizability at the interface of the passive film and the solution (*α* = 0.5) [34,35] and *B* = constant. Further, the *L_ss_* can be calculated using this relationship:(10)$$L_{ss}=\frac{\varepsilon \varepsilon_{0}A}{C_{eff}}$$
where *A* = surface area, *C_eff_* = effective capacitance measured from the EIS results after the PSP test according to Brug’s formula [36]:(11)$$C_{eff}={Q_{f}}^{\frac{1}{n}}\times {\left(\;R_{f}^{-1}\right)}^{\frac{n-1}{n}}$$

Hence, the film growth kinetics can be manifested as follows [37]:(12)$$\frac{dL_{ss}}{dt}=J\frac{\Omega }{N_{A}}$$
where *N_A_* = Avogadro’s number, Ω = molar volume per cation, and *J* = flux of the vacancies expressed by the first Fick’s law, as the sum of the two components: concentration gradient (*J_C_*) and potential gradient (*J_P_*), by:(13)$$J=J_{C}+J_{p}$$

The flux of oxygen vacancies (*J_o_*) under the influence of a concentration gradient and an electric field between the two interfaces of the passive film, can be expressed as:(14)$$J_{o}=-D_{o}\;\frac{\partial C_{o}}{\partial x}-2KD_{o}C_{o}$$

In a static condition, *J_C_* is considered as a constant. Hence, only *J_P_* can modify *J*. Since the current flow is a result of the donor–vacancy flow, the relationship between *J_P_* and the steady-state passive current (for double-charged oxygen vacancies) can be expressed as [33]:(15)$$J_{P}=-\frac{i_{ss}}{2e}$$

## 3. Results

### 3.1. Observation of Passivation Process

Figure 1a presents the anodic part of PDP curves, on which a common potential of a mid-plateau of passivation (0 V/SCE) for the three TiHfZrNb specimens is indicated and chosen for the PSP tests. The sudden change in the current density observed on PSP curves (Figure 1b) with small disturbances in the stable domain indicates a formation of passive film on the specimens with several processes that took place simultaneously [38,39]. The lowest current density of the steady-state plateau is observed for the 0.2 Nb specimen at 2.5 × 10^−7^ A·cm^−2^, while the highest was observed for the 0.4 Nb at 6 × 10^−7^ A·cm^−2^ (Figure 1b).

#### 3.1.1. Current Transient Analysis Using HFM

Here, the current flow at the surface generated from the process of nucleation and growth of the film was measured (Figure 1b and Figure 2a). The HFM carries the idea that all the current generated at the surface is consumed at the level of the film growth, without considering the dissolution process. The graphical representation of Equation (2), shown as the log *(i)* vs. log *(t)* plot (Figure 2b), indicates a linear relationship in the time range of 0.3 and 10 s for all specimens.

In addition, there is a clear deviation from the model at the early (t < 1 s) and later (t > 10 s) stages. The graphical representation of Equation (3) (Figure 2c) shows a similar deviation for the log *(i)* vs. *q*^−1^ plot at the early stage (t < 1 s). A linear increase in the plots that follow the model presented in Equation (3) is only valid at longer times. The passivation index and the *cBV* parameter, which effectively measure the passivation rate [21] are derived from the linear slope of the plots. As can be seen in Table 1, the increase in the Nb content led to an increase in both parameters during the early time.

#### 3.1.2. Current Transient Analysis Using DGM

The DGM approach uses the concept of the surface coverage fraction according to the Avrami kinetics, which defines several geometries related to the growth mechanism of the passive film. In the present work, a geometrical parameter, *n* = 1 was used, which is associated with a 2D growth mechanism with a constant thickness. Equation (4) was fitted (R^2^ = 0.99), and the parameters were derived at the early stage of all HEA specimens (Figure 3, Table 2). The passive film formation rates (*k*) increase as the Nb content increases (Table 2). This and other parameters are then used to calculate *i_diss_*, *i_film_*, and θ as a function of time presented graphically in Figure 3.

At a very early stage, we observed the coexistence of two mechanisms—dissolution and film formation—for all samples, as indicated by the parameters *i_diss_* and *i_film_*, respectively. For both the 0.3 Nb and 0.4 Nb samples, there is a rapid decrease in *idiss* and a corresponding increase in *i_film_*, until both parameters become equal, signifying equal kinetics. Following this point, *i_film_* continues to increase while *i_diss_* decreases, signaling the dominance of the film formation mechanism. This trend persists until the peak of the curve is reached, after which a decrease in *i_film_* suggests a transition from film formation to film growth. In contrast, *i_diss_* remains present in the 0.2 Nb sample throughout the test period. The fraction of the passive film coverage (θ) increases rapidly to θ = 1 after 1 s and 2 s for 0.4 Nb and 0.3 Nb, respectively. While for 0.2 Nb, the value of θ after 2 s is 0.05.

The passive film thickness was estimated from the PSP data based on the following equations:(16)$$i_{film}=\theta At^{-b}$$
(17)$$\theta =1-\exp\left(-kt^{n}\right)$$

When θ = 1, which means the first formed oxide layer, we can follow the evolution and growth of the passive film developed on the surfaces of 0.3 Nb and 0.4 Nb by the integration of *i_film_* as a function of time to determine the charge density of the film according to the following relation [40]:(18)$$q_{film}\left(t\right)=\int_{t=0}^{t}i_{film}\;dt$$

As seen in Figure 4b, the charge density for 0.4 Nb is higher than that calculated for 0.3 Nb, which is associated with the higher current density generated throughout the formation and thickening of the passive film. In Figure 4a, we see that the *i_film_* plots are divided into two domains, the first domain corresponds to the formation of the film presented by an intense rise in the current density up to a point that represents a total covering of the entire surface by the film. The second domain corresponds to the thickening of the film presented by a progressive drop in the current densities.

Thereafter, the thickness of the film is related to the charge density by the following equation [40]:(19)$$h\left(t\right)=q_{film}.\;\frac{M}{z\rho F}$$
where z = cation charge, M = molecular weight of the film, ρ = film density, and *F* = Faraday constant. From the XPS results obtained in the previous work [14], the oxide film was predominated by more than 65 at.% of ZrO_2_ and HfO_2_. An average value of the properties of the two oxides was used for the thickness calculation, *z* = 4, ρ = 7.68 g.cm^−3^, *M* = 166.6 g.mol^−1^, and *F* = 96,485.34 C.mol^−1^. Therefore, the estimated thickness of the passive film after 1 h of the polarization test is 2 nm for 0.4 Nb and 1 nm for 0.3 Nb, according to the fitting models presented in Figure 5.

### 3.2. Diffusivity of Point Defects in the Passive Film

#### 3.2.1. PSP at Different Potentials

Figure 6 displays the PSP plots under four different potentials for 1 h, showing the same behavior as that observed in the results of Figure 1b, which is a rapid drop in the current density, pursued by a stabilization in the form of a plateau, with the presence of small disturbances in the stationary domain. High quasi-stationary current densities are attributed to high potentials. Also, the starting current density peaks increase with the increasing applied potential.

#### 3.2.2. EIS after PSP

The resistivity of the passive film formed was investigated by the EIS technique at selected potentials in the passive region after PSP. Representative Nyquist plots of the three alloys are shown in Figure 7, depicting a truncated semicircle that characterizes the passive electrode/electrolyte interface [41]. A slight enlargement of the diameter of these semicircles was observed as the applied potential increased. However, a greater difference in diameter was revealed as the amount of Nb increased. A depressed loop in the high-frequency region indicates a phenomenon of dispersion frequency resulting the non-homogeneity of the surface. Nevertheless, the low-frequency arm is not observed to terminate at the real axis in this case because of the very high value of the polarization resistance. Consulting Bode diagrams (Figure 7e–g), the inverse phase angle vs. log (frequency) plot, which is more sensible to applied potential, indicates the resistivity behavior of the interface in the high frequency region (100–10 kHz) as well as in low frequency (0.19–0.01 Hz). In addition, the size of the Gaussian of the phase angle in the medium frequency region, when the charge transfer occurs at the interface, is characterized by a large size of more than three decades, suggesting that the passivation phenomenon held at the interface of all alloys with more than single time constant phenomena.

Based on this description, the impedance behavior of the alloy/passive film interface to the passive film/electrolyte interface was analyzed using the most appropriate electrical equivalent circuit [42], which constitutes an uncompensated solution resistance (*R_s_*), a passive film resistance (*R_f_*), a charge transfer resistance (*R_ct_*), a constant phase element for each the passive film (*CPE_f_*) and the double layer (*CPE_dl_*). Table 3 regrouped the extracting parameters obtained using the electrical equivalent circuit in Figure 7g. The CPE impedance is expressed as:(20)$$Z_{CPE}={\left[Q{\left(j\omega \right)}^{n}\right]}^{-1}$$
where *Q* (*CPE_f_* and *CPE_dl_* in Table 3) = magnitude that represents a pre-factor of *CPE*, and *n* = exponent of *CPE* (−1 ≤ n ≤ 1) that represents a parameter to measure the deviation from the ideal capacitive behavior of the film.

The impedance in the low-frequency range of EIS is approximately equivalent to the polarization resistance (*R_p_*), which equals the sum of *R_ct_* and *R_f_*. Here, the polarization resistance can represent the film resistance, correlating inversely with the corrosion rate. The *R_p_* values increased with the Nb content and the applied current. The highest *R_p_* value was obtained with 0.4 Nb at a passivation potential of 0.05 V. There are no significant differences in the *CPE_f_* values for the three HEAs with respect to the applied potential. This can be the result of the formation of compact oxide films on the metal surfaces at passivation potentials. However, the *CPE_dl_* values are noticeably affected not only by the Nb content but also by the applied potential. This indicates a modification in the rearrangement of charge species in the double layer, which is confirmed by the lower *n_dl_* value observed in 0.2 Nb.

#### 3.2.3. Semiconductor Properties of the Passive Film

Figure 8 presents Mott–Schottky plots for the passive films formed on the specimens at four different potentials. A positive slope, in the potential range of −0.1 to 0.05 V/SCE, indicates an n-type semiconductor behavior. A shift toward low values of Cs^2^ by increasing the polarization potential is obvious for all specimens. The imperfect linear domain in the Mott–Schottky plots indicates an inhomogeneous capacitance dispersion due to a non-uniform distribution of donors in the thickness direction. The donor densities of the film formed on the three specimens (see Figure 9a) can be calculated using:(21)$$\frac{1}{C_{s}^{2}}=\frac{2}{\varepsilon \;\varepsilon_{0}\;e\;N_{D}}(E-E_{FB}-\frac{kT}{e})$$
where *ε*_0_ = vacuum dielectric constant (8.854 × 10^−14^ F.cm^−1^), *ε* = dielectric constant (equal to 11.2, 11.1, and 11.3 for 0.2 Nb, 0.3 Nb and 0.4 Nb, respectively, calculated according to the oxides fraction in all specimens) [43], *N_D_* = donor density in the passive film, *E* = film formation potential, *E_FB_* = flat band potential, *k* = Boltzmann constant (1.38 × 10^−23^ J.K^−1^), and *T* = absolute temperature.

Figure 9 regroups the different representations of the parameters according to the film formation potentials. Defect densities (in the order of ×10^21^ cm^−3^) for all specimens decrease with the increase in the film-forming potential, with the lowest values for the 0.4 Nb specimen. The fitting of these results led to the determination of the constants, *w*_1_ and *w*_2_, as presented in Figure 9a. The models showed the defect density largely drops with an increasing Nb content under different potentials. The variation of film thickness as a function of potential follows a positive linear slope for all specimens, with an increase in the slope from 0.2 Nb to 0.4 Nb. From the fitting of Figure 9b, the *ε_l_* values for the 0.2 Nb, 0.3 Nb, and 0.4 Nb specimens are 15.1, 5.4, and 4.5 ×10^8^ V.m^−1^, respectively. The value of *D*_0_ (in the order of ×10^−23^ cm^2^.s^−1^) also increases with the potential and decreases with increasing the Nb content, as presented in Figure 9c.

As the film is made of a mixture of several oxides, there may be several types of oxides present, with the majority being Nb-containing ZrO_2_ and HfO_2_. Other additional oxides may also be present. For example, a study on Nb-doped ZrO_2_ discovered the presence of a small quantity of ternary oxide of Nb_2_Zr_6_O_17_ (ICSD card No. 01-072-1745) [44]. Thus, the reduction of the diffusivity due to the Nb addition can be viewed as a collective effect of Nb doping toward these oxides, and this feature is consistent with the previous experimental and theoretical works. The result, for instance, aggresses quite well with the result of a theoretical plus experimental study showing that the Nb substitution is highly effective in suppressing the oxygen vacancy concentration despite the corresponding increase in free electrons enabled by Zr-based defects since the oxygen vacancy concentration is the key mediator as a relatively fast oxygen diffusion. The same study also found the Nb enrichment near the surface relative to the one within the bulk. Other experimental works have also supported the conclusion, showing that even a small addition (a few %) of Nb alloys within many commercial Zr-rich alloys, such as Zirlo, M5, and E110, indeed results in a notable increase in their oxidation resistance [45]. As for the effect of Hf, the Nb^5+^ doping to the Hf^4+^-cation site has also been shown to significantly reduce the oxygen vacancies using a similar doping scenario [46]. Thus, overall, the results of the present work are consistent with the previous results of studies involving individual oxide phases. The same trend was observed for the *J_p_* flux (in the order of ×10^13^ vacancies per cm^2^.s) in Figure 9d.

## 4. Discussion

The formation of passive film on TiHfZrNb alloys occurred through two stages of passivation. First, a high kinetics stage led to the formation of the first oxide layer. As demonstrated in our previous work, the Nb content is higher on the outer side and lower on the inner side of the passive film [14]. The low potential drop promotes ion displacement to form a uniform film over the entire metal surface, which is characterized by a large fraction of Nb oxides. Second, a low kinetics stage leads to the growth of the passive film towards the metal substrate, which is characterized by the formation of a small fraction of Nb oxides in the inner side layer and the evolution of the whole film into a more resistive character. At the very early time of the passivation process, the two stages of dissolution and passive film formation coexist.

### 4.1. Kinetics of the Passivation Process

The present work focused on studying and following the development of the passivation process and the formation of passive film from the first oxide layer formed on a bare metal surface prepared by electrochemical dissolution of the old existing layer. This part of the analysis will be based on current transient analysis using the two theoretical and mathematical approaches: the HFM and the DGM.

According to the current transient analysis using the HFM, the film formed on the 0.4 Nb specimen has the highest passivation rate and *cBV* value. Passivation index values of 0 to 0.5 indicate the formation of an imperfect dense film that contains defects and vacancies, probably as a result of the simultaneous dissolution and passivation processes, whilst a passivation index that approaches unity (1) indicates the formation of a dense and compact film [47]. The *cBV* is linked to a potential drop across the film or a potential difference between two sides of the film (the metal and the solution sides), representing the driving force for ion mobility and conduction. An increase in the *cBV* value is linked to a decrease in the barrier energy (activation energy) for the ion displacement across the film, hence reducing the film density and its protective characteristics [28,40]. However, based on an assumption of the 100% yield of the HFM theory, a low-energy barrier will facilitate the displacement of ions for the film formation and accelerate the kinetics of passivation and compensation of defects, which agrees with the high passivation index of the 0.4 Nb specimen. This trend indicates that Nb improves the passivation kinetics. In addition, the passivation process is a consequence of the two stages of growth and thickening mechanisms of the passive film. The first stage informs that the film is less resistant to the flow of a current by ionic conduction. While during the second stage, the formed film is more resistant [19]. However, this does not necessarily imply that the mechanism is governed by the formation of two separate films. Rather, the whole film may evolve altogether when the growth kinetics of the film transition from one regime to another [19]. The HFM has a limitation that comes from the assumption that the current density generated at the bare surface during the early time phase is used in totality for the formation of the passive film and its growth, neglecting in totality any minimal contribution of the anodic dissolution process. This assumption was explained by the formation of the first oxide layer that blocks the ion movement under the effect of a powerful electric field and consequently limits the metallic dissolution [19]. Nevertheless, this approach explains the influence of the increasing Nb content in limiting the anodic dissolution by the concurrent formation of a uniform film, which is attributed to the rapid passivation kinetics.

Based on the current transient analysis using the DGM, the results indicate that at a very early time (before *i_diss_ = i_film_*), the anodic dissolution process is the most dominant for 0.4 Nb and 0.3 Nb specimens. Later, as *i_diss_* quickly falls below *i_film_*, the passive film formation process and the subsequent thickening dominate the passivation process. The surface coverage fraction increases very rapidly up to θ = 1, signifying that the formation of the first uniform oxide layer over the entire surface is reached after 1 s for 0.4 Nb and 2 s for 0.3 Nb under passive polarization. Unlikely, the 0.2 Nb exhibits a permanent dissolution process and a low surface coverage during the early time, which implies a continuous anodic dissolution of metal due to a low non-uniform film formation. Using the PSP data (Figure 1b and Figure 2a), the current transient analysis of the DGM theory provides a film growth rate and a film thickness estimation as a function of time by calculating the charge densities generated during the process. This model also allows a quantitative estimation of the film thickness variation, or in other words, a follow-up on the thickening process of the formed film. The formed passive film on 0.4 Nb is thicker than that on 0.3 Nb since the thickness of the film is linked to its passivation rate. The thickness estimation can be qualitatively examined from the plots of current and charge densities linked to the film formation and growth processes. Hence, for a given time interval (∆*t*), *i_film_* (0.4 Nb) > *i_film_* (0.3 Nb), and therefore *q_film_* (0.4 Nb) > *q_film_* (0.3 Nb), as presented in Figure 4.

### 4.2. Diffusivity of the Point Defects

The PDM theory estimates the diffusivity of point defects in the passive film and gives a microscopic description of the growth in both transient and steady states of passivation. It also provides an analytical expression for the concentration and flux of defects in the passive film, offering a possibility of quantitative analysis.

The plots of PSP at different potentials (Figure 6) indicate that passive film was formed on all specimens, as expressed by the rapid decrease in current density over time from a transient state with very high current density to a steady state where a static balance is maintained. Based on the PDM theory, the steady-state current density (*i_ss_*) for a passive film is almost independent of the applied potential (d*i_ss_*/d*E* = 0), which can be explained by the equalization effect of the oxidation degree of a metal atom to its dissolution degree [25]. During the PSP tests, the passive film was grown at each potential for 1 h to ensure that the system reached a stable state. In the course of passive film formation, no obvious current peak was observed, indicating a good resistance of the film to pitting and giving evidence that *i_ss_* is independent of the film formation potential, which is consistent with the PDM theory. The *i_ss_* is a key parameter to illustrate the kinetics of film growth and dissolution, where a small difference in *i_ss_* as a function of potential indicates a change in the film microstructure [48,49]. Moreover, the decreasing number of donor density (*N_D_*) with increasing film-forming potential (Figure 9a) is also consistent with the PDM prediction that expects a passive film growth by generating oxygen vacancies at the alloy/film interface and by their annihilation at the film/solution interface [23].

The EIS results showed an unbroken passive film is formed at potentials exceeding the Flade potential. Indeed, an enhancement in the values of *R_p_* and *R_ct_* with the addition of Nb. The corrosion resistance of alloys is closely related to the passivation state of the surface of materials. Thus, the influence of Nb on the passivation process was the main reason for the increase in corrosion resistance. Generally, anodic dissolution, passive film formation, and chemical dissolution of the passive film coexist during the passivation process on the electrode surfaces in Hank’s simulated body fluid, originating from localized corrosion caused by electroactive ions present in the solution. From the Bode plot, it appears that the phase angles and impedance modulus of all alloys are quite similar in size and exhibit the typical features of passive alloys with phase angles close to −90° over a wide frequency range [50]. It is reasonable to deduce that the electrode/electrolyte interface of all alloys undergoes similar electrochemical processes.

The semiconductor behavior of a passive film is strongly n-type semiconductors (Figure 8) with interstitial cations and oxygen vacancies as the major dopants. The donor concentration in the formed film decreases by increasing the Nb content in the alloys. Meanwhile, the *N_D_* decreases with increasing the potential for all specimens. The variation in the Mott–Schottky plots at relatively low potentials (at the beginning of plots) for each applied potential may be associated with the electron depletion effect near the electrolyte/film interface, which occurs during the rapid increase in the applied potential for the first time that creates a region of uniform donor density [33]. Moreover, the change in *J_o_* flux with increasing potential can be attributed to a change in the film microstructure [48,49]. An example of anodic films on titanium showed that for low anodic film formation potentials, the formed film is relatively amorphous due to the presence of TiO, TiO_2,_ and Ti_2_O_3_, while for high film formation potentials, the structure becomes more crystalline due to the presence of TiO_2_ [48,49].

### 4.3. Proposed Passivation Mechanism

A graphic illustration (Figure 10) is proposed to describe the film formation (passivation) mechanism. According to the PDM, the film growth process is based on the flux of vacancies across the passive film, which for an n-type film are the oxygen vacancies and/or the cationic interstitials that act as donors. During the diffusivity determination, it is difficult to differentiate the contribution of oxygen-ion vacancies and cationic interstitials to the mechanism, which implies a consideration based on the combination of these two types of point defects. In Hanks’ simulated body fluid, the presence of Cl^−^ ions can cause pitting on the passive film. The adsorption of the highly negative charge Cl^−^ ions is very favorable on a passivated surface having an n-type characteristic due to the presence of highly positive charge oxygen vacancies. At the metal/film interface, oxygen vacancies and cationic interstitials are produced and then moved to the film/electrolyte interface to be consumed. Hence, the breakdown resistance of the passive film depends on the kinetics of the vacancy generation and annihilation reactions [23]. The low surface fraction coverage by the passive film on the 0.2 Nb specimen is attributed to the high defect concentration (high *D*_0_) generated from an autocatalytic reaction due to the Cl^−^ ions adsorption onto the film. This implies an increase in oxygen vacancies and cationic interstitial mobility through the film as the result of a high electric field strength associated with a very thin film [51]. Arriving at the film/electrolyte interface, they react with Cl^−^ to form vacancy condensates and, subsequently, initiate active dissolution sites. Therefore, the dissolution mechanism with a high dissolution rate will be dominant over the film formation and growth mechanism. Higher Nb content promotes the formation of a first compact passive barrier covering the entire surface rich in Nb oxides, which reduces the point defects concentration and consequently limits the adsorption of Cl^−^ ions. Hence, the film growth mechanism dominates the passivation process and promotes a film thickening, as shown in Figure 3c and Figure 5, to form a highly resistant film rich in oxides of Zr, Ti, and Hf in the inner layer and in Nb oxide and hydroxide in the outer layer [14].

In the context of implant materials, some clinical cases of implant failure due to corrosion are still reported on the current medical-grade alloys [52,53]. The routine corrosion tests that follow the applicable standards, like the ASTM F2129 [54], are unable to reveal the detailed corrosion mechanism. The present work reveals a detailed role of Nb in the corrosion mechanism of TiHfZrNb HEAs in the aggressive Hanks’ simulated body fluid. It confirms the high resistance to corrosion/dissolution and rapid repassivation ability of TiHfZrNb_0.3_ and TiHfZrNb_0.4_ alloys, warranting further in vitro biocompatibility testing to comprehensively confirm their potential use as new biomedical implant materials.

## 5. Conclusions

The present work reveals that niobium plays a key role in improving the kinetics of the passive film formation on TiHfZrNb high-entropy alloys. The increase in the niobium content in the alloys (from 0.2 to 0.4 molar ratio) accelerates the formation kinetics of the first barrier layer with a rapid total surface coverage, promotes the dominance of the passive film formation and growth mechanism while limiting the film dissolution mechanism. Higher niobium content also reinforces the capacitive resistant behavior of the protective barrier by enrichment with niobium oxides and reduces the point defect density and their mobility within the film. With the high resistance to dissolution and rapid repassivation ability in the aggressive Hanks’ simulated body fluid, the TiHfZrNb_0.3_ and TiHfZrNb_0.4_ alloys confirm their great potential as new materials for biomedical implants and warrant further biocompatibility testing.

## Figures and Tables

**Figure 1 jfb-15-00305-f001:**
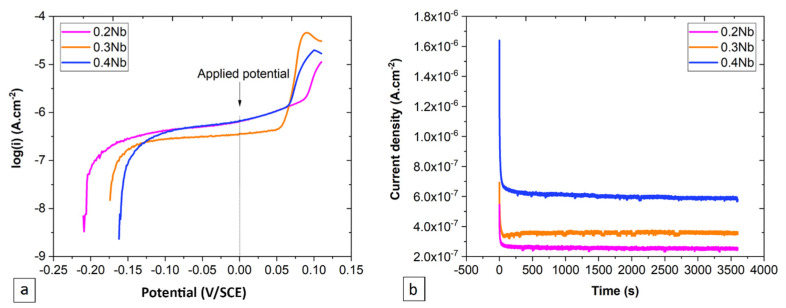
Polarization-test results are plotted as (**a**) the anodic part of PDP graphs indicating the selected potential for the PSP test, and (**b**) PSP graphs at the selected potential (*E_pass_* = 0 V/SCE).

**Figure 2 jfb-15-00305-f002:**
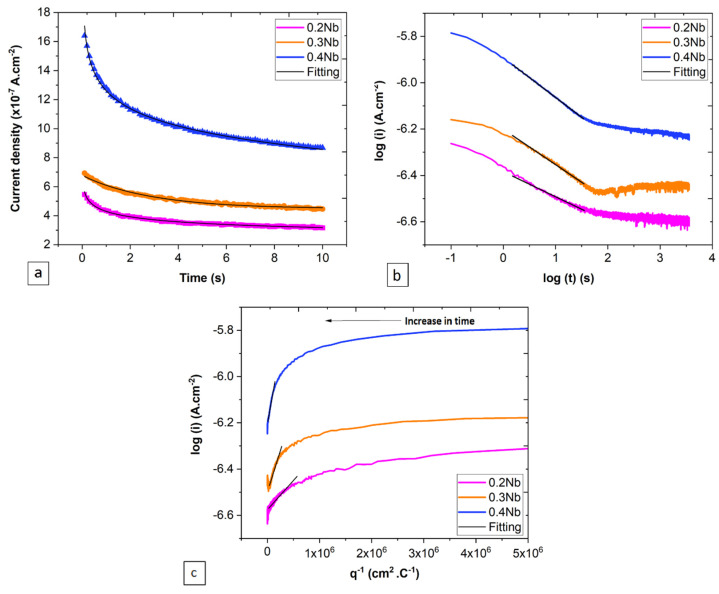
Measured current density during the early time of the PSP tests, represented as (**a**) current density vs. time plot, (**b**) logarithmic plot of current density vs. time, and (**c**) logarithmic plot of current density vs. inverse surface charge density.

**Figure 3 jfb-15-00305-f003:**
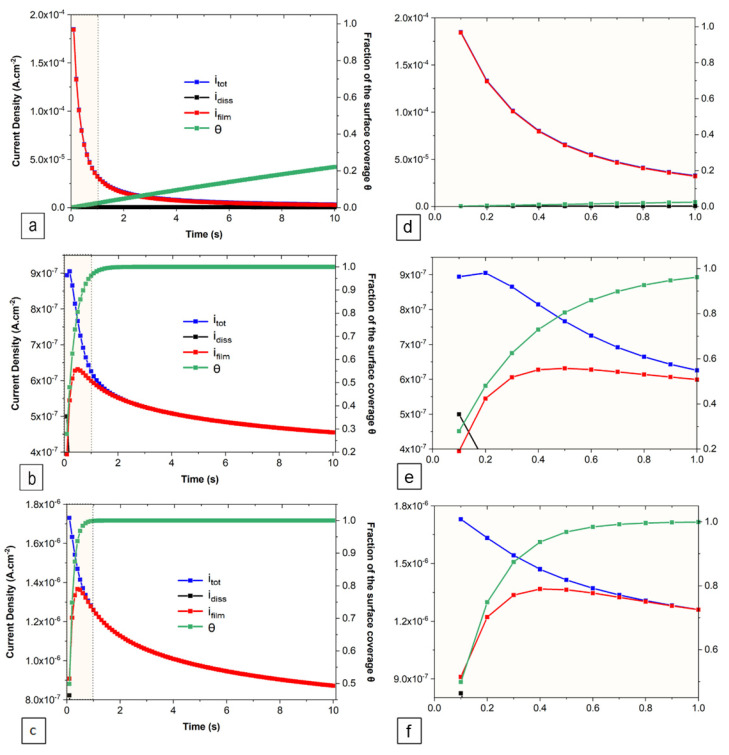
Representation of *i_diss_*, *i_film_*, and θ as a function of time for (**a**) 0.2 Nb, (**b**) 0.3 Nb, (**c**) 0.4 Nb, and their zoomed sections (**d**,**e**,**f**, respectively).

**Figure 4 jfb-15-00305-f004:**
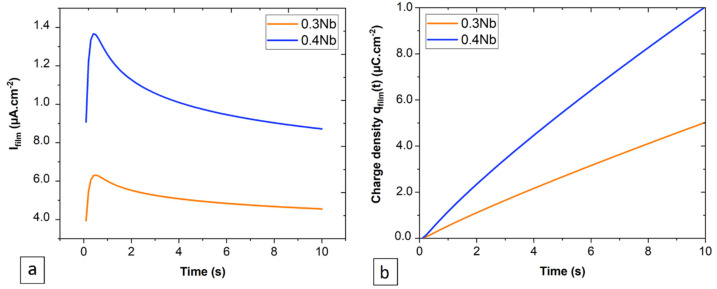
Film formation process on 0.3 Nb and 0.4 Nb specimens represented as (**a**) current density vs. time and (**b**) surface charge density vs. time.

**Figure 5 jfb-15-00305-f005:**
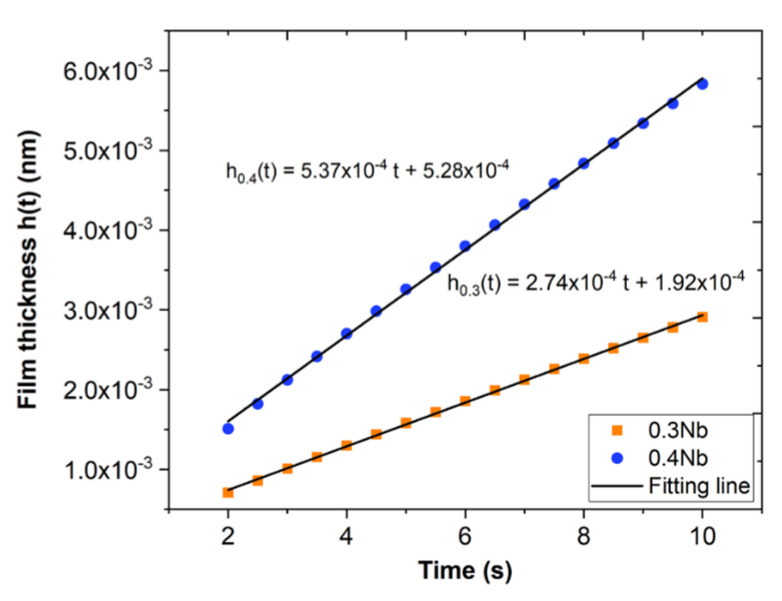
Plots of film growth as a function of time (R^2^ = 0.99 for both models).

**Figure 6 jfb-15-00305-f006:**
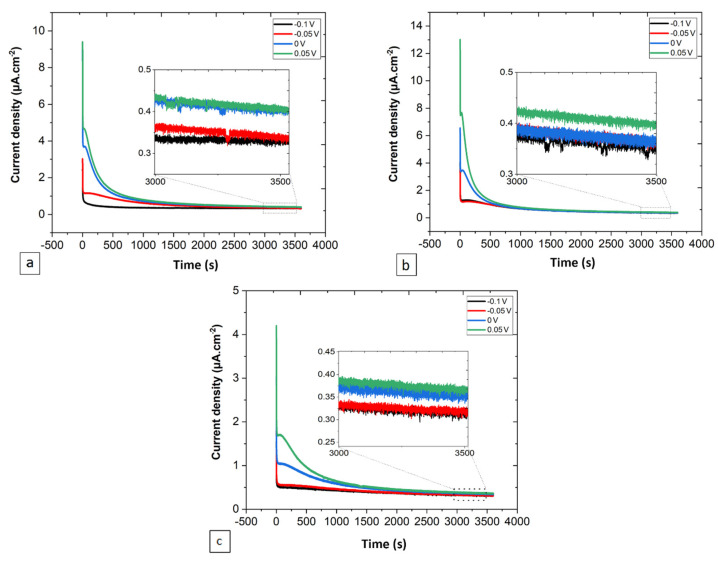
Representation of PSP test results as current density vs. time for (**a**) 0.2 Nb, (**b**) 0.3 Nb, and (**c**) 0.4 Nb under four different potentials: −0.1, −0.05, 0, and 0.05 V/SCE.

**Figure 7 jfb-15-00305-f007:**
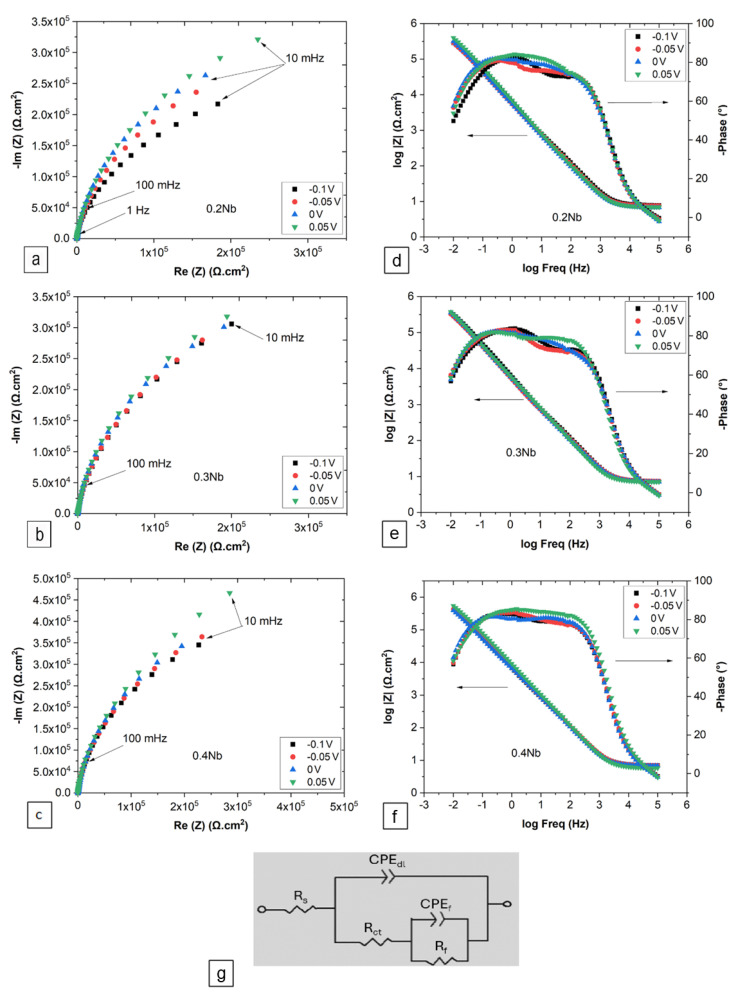
EIS results represented as Nyquist and Bode plots for (**a**,**d**) 0.2 Nb, (**b**,**e**) 0.3 Nb, (**c**,**f**) 0.4 Nb, and (**g**) equivalent circuit, after being subjected to PSP at four different potentials.

**Figure 8 jfb-15-00305-f008:**
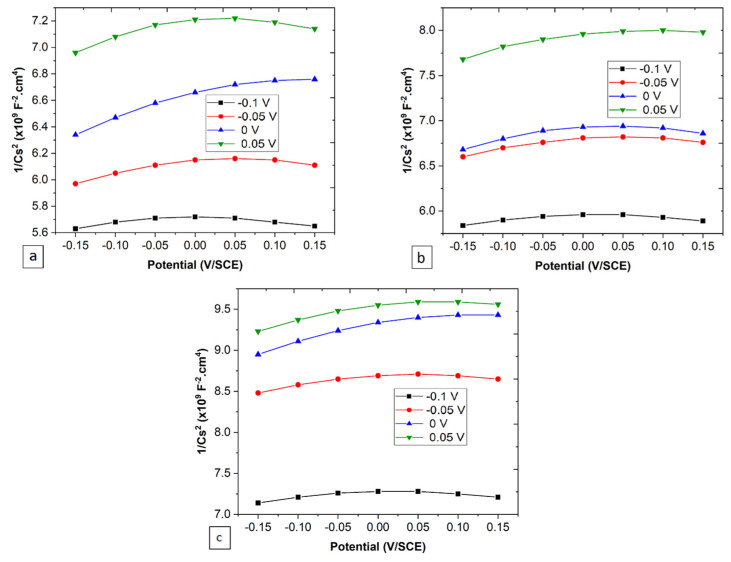
Mott–Schottky plots for (**a**) 0.2 Nb, (**b**) 0.3 Nb, and (**c**) 0.4 Nb under four different potentials.

**Figure 9 jfb-15-00305-f009:**
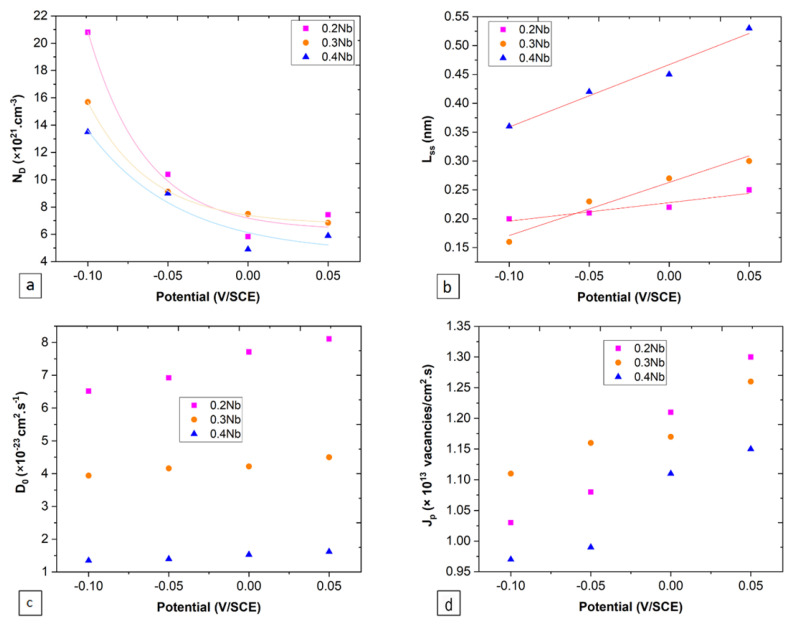
Plots of semi-conductive properties of the passive films formed on the alloy specimens at different potentials, showing: (**a**) donor density (*N_D_*), (**b**) steady-state passive film thickness (*L_ss_*), (**c**) vacancy diffusion coefficient (*D*_0_), and (**d**) vacancy flux (*J_p_*).

**Figure 10 jfb-15-00305-f010:**
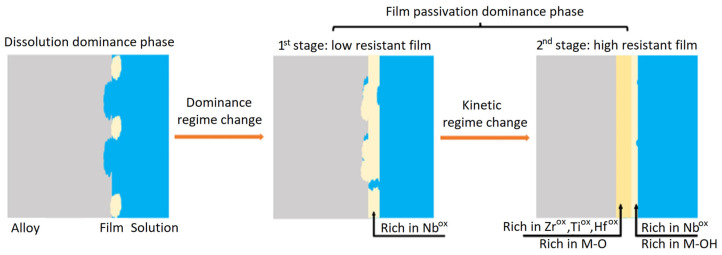
Schematic illustration of the film formation mechanism, showing dissolution dominance phase followed by two stages of film passivation dominance phase.

**Table 1 jfb-15-00305-t001:** Passivation index and *cBV* parameters.

Specimen	Passivation Index	cBV (×10^−7^ C.cm^−2^)
0.2 Nb	0.12	2.53
0.3 Nb	0.14	7.18
0.4 Nb	0.17	12.66

**Table 2 jfb-15-00305-t002:** Fitting parameters derived from the plots in Figure 3.

Specimen	*i_peak_* (×10^−7^ A·cm^−2^)	*k* (s^−1^)	*A* (×10^−6^ C.cm^−2^)	*b*	R^2^
0.2 Nb	5.65	0.025 ± 0.07	32.2 ± 10.1	1.06 ± 0.0026	0.99
0.3 Nb	6.93	3.27 ± 1.18	0.60 ± 0.00162	0.12 ± 0.0016	0.99
0.4 Nb	16.4	6.90 ± 3.37	1.26 ± 0.00166	0.16 ± 0.0008	0.99

**Table 3 jfb-15-00305-t003:** Fitting parameters obtained from the plots in Figure 7.

Specimen	Potential	*R_s_* (Ω.cm^2^)	*CPE_dl_* (×10^−6^ F.s^(n−1)^)	*n_dl_*	*R_ct_* (Ω.cm^2^)	*CPE_f_* (×10^−6^ F.s^(n−1)^)	*n_f_*	*R_f_* (×10^6^ Ω.cm^2^)
0.4 Nb	0.05 V	6.09	7.41	0.93	3611	3.94	0.9	1.30
	0 V	6.75	8.65	0.93	3426	3.78	0.94	0.97
	−0.05 V	7.06	9.24	0.93	3230	3.85	0.93	0.95
	−0.1 V	7.12	10.45	0.93	2954	3.95	0.93	0.93
0.3 Nb	0.05 V	7.12	13.94	0.92	2720	2.95	0.94	0.92
	0 V	7.08	15.16	0.92	2538	3.89	0.95	0.84
	−0.05 V	7.01	17.79	0.92	2050	4.71	0.94	0.80
	−0.1 V	6.73	29.43	0.9	774	5.03	0.92	0.79
0.2 Nb	0.05 V	7.5	17.63	0.91	957	5.9	0.98	0.79
	0 V	7.8	22.08	0.9	826	5.57	0.97	0.79
	−0.05 V	7.56	28.06	0.88	774	3.21	0.97	0.72
	−0.1 V	7.8	34.87	0.86	692	3.09	0.9	0.58

## Data Availability

The original contributions presented in the study are included in the article, further inquiries can be directed to the corresponding author.

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
