# Peer review of "Role of Niobium on the Passivation Mechanisms of TiHfZrNb High-Entropy Alloys in Hanks’ Simulated Body Fluid"

_jfb, 2024, doi:10.3390/jfb15100305_

Round 1
Reviewer 1 Report
Comments and Suggestions for Authors
This paper (jfb-3231607) evaluates the role of niobium on pas-sivation kinetics and electrochemical characteristics of passive film on TiHfZrNb alloys formed in Hanks simulated body fluid by analyzing electrochemical data with three analytical models. Results confirm that higher niobium content in the alloys reinforces compactness of the passive film by favoring the dominance of film-formation and thickening mechanism over dissolution mechanism. Overall, this work the manuscript is meaningful and worthy of publication. However, some issues need further clarification.
Detailed:
1. Supplementing the surface morphology and composition after potentiostatic polarization (PSP) can verify your conclusion more intuitively.
2. Fig. 7h, the authors selected the R(Q(Rct(QRf))) as the equivalent-circuit models, why not adopted the R(Q(Rf(QRct))) model.
3. Fig. 10, I suggest the film-formation mechanism diagram of three compositions of TiZrHfNbx (with x = 0.2, 0.3, and 0.4 in molar ratio), which can better emphasize the difference of passivation film formation.
4. Hafnium usually does not have any impact on human health, but some compounds of hafnium (such as hafnium chloride) are toxic and can have adverse effects on human health, which can corrode the skin and mucous membranes. Is it suitable as a new material for biomedical implants?
Comments on the Quality of English LanguageMinor editing of English language required.
Reviewer 2 Report
Comments and Suggestions for Authors
The manuscript "Role of Niobium on the Passivation Mechanisms of TiHfZrNb High entropy Alloys in Hanks Simulated Body Fluid " is well written, and the English is good.
The description of the processes is intricate and technical, suitable for a specialised audience. Nonetheless, certain terminology (e.g., "arc-melting process") may be briefly elucidated for clarity without compromising technical precision. The document might be enhanced by incorporating further interpretive analysis of the implications of the data. Although it comprehensively covers the data, it would improve by emphasising key points more strongly.
The phrase in line 39, "The multi-element concept of HEAs offers extensive options for selecting non-toxic elements to guarantee biocompatibility..." can be revised for brevity. The diverse composition of HEAs allows the use of non-toxic materials to guarantee biocompatibility.
Line 32: "Prior approval by the regulatory organisations, implant materials are required to show a high resistance..." necessitates a comma following "organisations."
Lines 106-109: "The high-field ionic-conduction model posits that a passive film is generated and augmented due to the displacement and transfer of charges across the film, instigated by a strong electrical field..." This sentence may be deconstructed for enhanced clarity.
Lines 27-28: The citation “[12]” lacks clarity. The reference must include the author and year or denote "In Press" for forthcoming citations.
Line 34: The citation “[34]” lacks clarity; ensuring that references are appropriately linked to the appropriate sources is imperative.
The term “previous work” is referenced (Lines 47, 105, 366), although it would be beneficial to provide explicit citations for those articles instead of only alluding to them.
I recommend a minor revision.
Reviewer 3 Report
Comments and Suggestions for Authors
The article entitled " Role of niobium on the passivation mechanisms of TiHfZrNb high-entropy alloys in Hanks simulated body fluid" is devoted to the study of the passivation kinetics and electrochemical characteristics of passive film on TiHfZrNb alloys. The novelty of this work is the determination of the predominant mechanisms of the passivation process of TiHfZrNb alloys. Also, the influence of composition of the TiHfZrNb alloys both on the kinetic of passivation process and the capacitive-resistant behavior of alloys have been discussed. The results of this work are emphasized the advantages of TiHfZrNb alloys with high niobium content which shown enhanced the passivation kinetics to rapidly form the first layer and total surface coverage, reinforces the capacitive-resistant behavior. From a practical perspective, this work is of interesting because some TiHfZrNb alloys have been regarded as new materials for biomedical implants. Overall, the article is well written and only minor corrections are needed.
1. The results illustrated on Fig. 3 are not fully correlated with the following observation " A decrease in idiss and an increase in ifilm is observed at a very early stage until the two parameters become equal after 106 ms for 0.4 Nb and 130 ms for 0.3 Nb". The behavior of idiss parameters should be better assign.
2. Previously, the corrosion tests have been performed for studied alloys (10.1016/j.electacta.2022.140651) at the same equivalent circuit model. So, the comparative analysis of electrochemical characteristics obtained here with ones obtained (10.1016/j.electacta.2022.140651) in should be done.
3. The significant differences in charge transfer resistance (Rct) values within series of samples are observed (see Table 3) in this work, but these values weren't differ on the two orders for these samples in the previous work (10.1016/j.electacta.2022.140651).Please justify.
Reviewer 4 Report
Comments and Suggestions for Authors
I'd suggest accepting the manuscript after minor revision.
In their work authors investigate quaternary TiHfZrNb alloy for biomedical application in terms of surface oxide film. This field is prospective and important since Ti-based alloys containing biocompatible alloying components appear to be promising materials for various biomedical devices. While structure and mechanical properties of such alloys determine their functional behavior it is surface structure and chemical state that defines biochemical compatibility of the material. In such way electrochemical behavior of these materials as well as their surface features should be under the spotlight.
The manuscript is well organized and the set of the selected methods is appropriate. The following issues list should be mentioned:
1) Lines 77-79: could you please specify if initial ingot had colored film pointing to a more oxidized surface? Also please note that after removing initial oxide state and obtaining mirror-like surface new natural oxide film forms immediately.
2) Chemical composition of the alloy is not clear. Authors increase Nb content from 20 to 40 at. % at the expense of even decrease of each other element? Does it mean that their content is equiatomic?
3) Could you please comment more on the quite noticeable difference between 0.2Nb and 0.3/0.4 Nb ifilm values and plots in Figure 3.
4) Could it be that the dominance of the film formation process increasing with Nb content increase is related to the lowest enthalpy formation of Nb2O5 (− 1771 * 10−3 kJ/mole) among all other oxides in the film?
5) Semiconductor properties testing compliment the work greatly. As authors mention that Nb can dope ZrO2 and HfO2 oxides it is natural to suggest that the reversed situation can occur as well. Would it deteriorate the film properties in terms of dissolution/growth balance neglecting that benefit?
6) Line 360 a small typo: “TiHfZrN”
7) The suggested film growth mechanism is clear and sound. Would you assume that in the case of film fracture (which is possible for medical devices) the film self-healing would follow the same two stages?
Comments on the Quality of English Language~
Round 2
Reviewer 1 Report
Comments and Suggestions for Authors
The author answered the related questions carefully, and the current version can be accepted.
Comments on the Quality of English Language
Minor editing of English language required.
Author Response
Comment: The author answered the related questions carefully, and the current version can be accepted.
Respond: We appreciate the reviewer's final comment and thank a lot for dedicating their time to improve our manuscript. We carefully reviewed the manuscript to further improve its English.